# Reflecting Culturally Responsive and Communicative Teaching (CRCT) through Partnership Commitment

**Zainal Berlian** [1,*] **and Miftachul Huda** [2,*]

1   Faculty of Tarbiyah (Education) and Teacher Training, Universitas Islam Negeri Raden Fatah, Palembang 30126, Indonesia
2   National Child Development Research Center (NCDRC), Faculty of Human Sciences, Sultan Idris Education University, Tanjung Malim 35900, Malaysia
*   Correspondence: zainalberlian68@radenfatah.ac.id (Z.B.); miftachul@fsk.upsi.edu.my (M.H.)

**Abstract:** The strategic approach to enhance cultural skills is aligned with social feeling and sense of belonging as an important element to underlie the interaction pathway with others' diverse backgrounds. Such an approach of teaching model could be transformed and prepared to encourage ability to communicate and relate to others from diverse backgrounds. The following phase of sufficient integration amongst cognitive, social and emotional substance is reflect in the culturally responsive and communicative teaching (CRCT). This mode is an important element to advance the diverse students amidst an increasingly complex and pluralistic society. This paper is aimed at examining CRCT by having a critical look into the partnership commitment amongst interracial teachers' daily school interaction. Data collection was conducted with 10 interracial teachers with a focus on the approach of how their partnership commitment is reflected and practiced in their daily school interaction. Thematic analysis was employed to obtain the main points of data to give insight into the multicultural school society. The findings reveal that an understanding of CRCT articulated into partnership engagement commitment has a core of three main points. These are: expanding self-awareness on cultural identity; enhancing culturally mediated emotion of social situations; and developing instructional practice for cultural sensitivity. The implication for students' experiential learning in the multicultural classroom was also discussed. The contribution of this paper can be exerted among those involved in the education sectors. These include students, educators, principals, learning facilitators, researchers, educational technologist, instructional designers and practitioners. This is also included among the researchers who are interested in exploring, understanding and developing discerning perspectives on diversity and diverse learners in 21st century education. This study is expected to contribute by giving solving skills with the strategic approach articulated into a particular guideline to give a clear understanding in responding the multicultural classroom in the interracial school society context. This study is considered to elaborate the good feedback on the importance of CRCT in responding to emerging trends on the facades of diversity among diverse learners.

**Keywords:** culturally responsive teaching; partnership engagement; commitment and intercultural teachers

## 1. Introduction

The issue of diversity has been widely discussed due to its expansion of diverse learners around the world. The raising of this issue in the classrooms indicates that diversity places new and important challenges on teacher education [1]. In this regard, the teaching model plays a crucial role to inspire and encourage the potential, with a commitment to excellence, of the students' ability of personal and social growth as a beneficial contribution through advice and guidance. As a result, it is crucial to empower the teacher's skills through professional training. The wide range of this important concern

includes a comprehensive focus on promoting personal and social awareness, empathy, collaborative skills and conflict literacy [2]. Moreover, it is also used in both teaching and learning skills and as the ability of an individual to understand and respect values, attitudes and beliefs. The arrangement process should consider and respond appropriately across cultural differences in planning, implementing and evaluating the educational process.

In addition, it is necessary to respond the challenges of diversity through adopting the teaching strategy in the multicultural classroom context. The extent of process included here is empowering a responsive and communicative approach in teaching performance. It would contribute to give a valuable essence to emphasize the way that can be carried out in multicultural school society context. An attempt to empower the teaching approach for diverse learners plays a significant role in underlying the way on how to integrate the skills from theory to applications [3]. Moreover, the promotional strategies to solve the multicultural classroom in the interracial school society context should be considered to expect to contribute good feedback. Among the significant points that can be seen are to provide insight on giving a clear understanding and insightful view on the importance of teaching approach for diverse learners [4]. Moreover, it also helps to determine the emerging trends on the facades of diversity with a particular guideline reflected in the teaching strategy [5]. As a result, this teaching approach should be engaged with in an attempt to lead to the significant engagement of diverse learners. Hence, the learners may have critical thinking and social awareness integrated into the extent of their experiential learning in the multicultural classroom.

In order to provide a strategic approach for developing social and self-awareness among diverse learners, a core set of congruent behaviors, attitudes, and policies need to be considered within the multicultural school society context. Since the social development of self-awareness is considered to be the teacher's duty, it is an important part of developing social feeling through exploring the teachers' social response in their daily school life [6]. The form of such an arrangement could be addressed though a cognitive, social and emotional basis for enhancing human development. Such an element is needed as an important pathway to prepare diverse students in the midst of increasingly complex and pluralistic society [7]. Towards the wide spectrum of ability and awareness, including cultural and physical ability, this would give insight into improving social development as the preparation in enabling the students to work in a global learning context with the skills of creative thinking and self-awareness.

Therefore, it is required for teachers to develop the ability to communicate with and relate to others from diverse backgrounds. The need to promote culturally responsive teaching (CRT) could be considered to be particularly aligned with a communicative basis in facilitating communication skills. As a result, the form of preparing such skills in the teaching approach might work effectively, not only for the partnership's diverse background but also for students from different backgrounds. In this regard, this paper aims to examine the culturally responsive and communicative teaching (CRCT) through dealing with the partnership engagement commitment reflected in their school's daily social interactions amongst interracial teachers.

## 2. Literature Review

### 2.1. Transforming CRT to CRCT Skill Approach

Culturally responsive teaching (CRT) is conceived as an aspect of teaching performance configured through an understanding between students' classroom practice and their home cultural lives [8]. The following practice is engaged with in the teaching skills approach generated as a bridge to perform CRT. The additional component refers to the way of knowledge construction to guide the students to perform their personal and cultural skills. Such practices could be employed in the curriculum context in order to promote learning and making the culture of the classroom inclusive of all students. With the varied assessment of practices, this initiative is entirely an effort to support the teaching skills

performed with an entire awareness towards the philosophical approach to the classrooms' contextual bases [9].

Addressing the outstanding component of the practical implementation of CRT, its integration into teaching skills with consciousness about the cultural identity and history of their students is a necessity. Attempts to bring the cultural background of community to the classroom are involved into such alignment processes of families in the educational process [10]. Thus, the necessity to bring the students' cultural skills into the classroom has to carry on helping the teachers to strengthen their CRT in the classroom. The ability to address the pedagogical use refers to enhancing and improving the understanding of the ways students learn [11]. The extent to which the teaching approach may implement culturally responsive teaching (CRT) could be proposed for displaying the cultural competence skills in the teaching within a cross-cultural setting. However, this attempt seems to lack the main instruction in terms of addressing their knowledge in their classes. This is because of the lack of communicative skills based on the particular approach in the way in which teachers and students may have a simultaneous interaction.

The need to develop comprehensive insight should be taken into consideration to stress the way that 'communicative skills' can support CRT. It is required to respond to the need of expanding the teaching approach with the significance of encouraging critical thinking on social and personal awareness among diverse students [12]. The significant regulation could be determined into the attempts to restructure CRT as the core element in building culturally responsive skills [13]. With this regard, responsive teaching needs to be associated with pedagogy grounded in the teachers' way of collaborating and working to relate course content to the cultural context to enable the student to gain communicative skills. To go beyond the attainment across a diverse cultural context, CRT model needs to further transform the communicative aspect. As such, it could be elaborated to mean culturally responsive and communicative teaching (CRCT). This aims to enhance the cultural awareness with communicational teaching skills within pedagogy to recognize the importance of students' cultural references in all aspects of learning [14].

CRT is the development model of culturally responsive teaching from the global perspective to communicate real meaning with natural strategies for the acquisition of real communication [15]. The communicative approach is based on comprising the plurality and equality covered within intellectual and moral development in the new paradigm of varying diversity. In particular, addressing management skills with adaptive teaching competencies, for instance, could be engaged into the application guidelines in supporting and assessing multiple channels to extract new insights of value of knowledge material [16]. As a result, the model of the CRCT approach could be appropriately expected, given the current trend of multicultural education contexts. Hence, it is significant to consider the manifestation of numerous challenges that need to be solved by the teachers to achieve a balance between the professional basis and cultural approach in their teaching style.

*2.2. Being Consistent in Partnership Commitment*

Attempts to construct consistent skills are arranged with social and emotional development in enabling the students to commit their learning and their life experience [17]. The expectation to comply with connecting the learning experience into the real-life circumstance is necessary to possess an emotionally supportive climate in the multicultural classroom [18]. It can be engaged with among teachers and students in the way that they can support their social and emotional development process. In this view, optimizing school rules with regulations to enable the learners' positive attitudes to skillfully connect their experiences is also important. This is because the essence of diverse unique experiences needs the capacity to recognize and manage emotions in terms of problem solving [19]. An effectively established positive relationship with others could be supported by such competencies oriented to their social and emotional development. In particular, the process of acquiring and effectively applying the knowledge, attitudes and skills is necessary to

recognize and manage emotions, combining behavior and emotion [20]. As a result, it enables the students to get a lot out of their learning preferences.

In particular, developing care and concern for others will create an underlying component to strengthen the learners with academic skills oriented towards the multicultural approach. It is necessary address one's responsibility in making decisions. As such, how to learn, practice and apply skills in the midst of society must be considered to discover how the learner can handle with engaging in positive activities in and out of the classroom. Further, establishing positive relationships to reach the initial skills empowers the enhanced and nuanced input integration obtained from multiple constituencies such as students, parents, teachers and community members [21]. The main point aims to address the increasingly complex situations to be learned by students and, further, to ensure their capability for social relationships and citizenship [22]. As a result, the approach to conduct oneself capably in handling challenging situations may enable all students to achieve social and emotional development in the multicultural classroom. Such instruction in this way, which should be incorporated with tolerance, is required to encourage the social engagement.

Getting to know the background of the global learning context around social awareness should involve engaging with it in depth [23]. Incorporating responsive-based self-awareness at the community level requires one to have the capacity to understand dynamics in terms of strengths, concerns, conflicts and challenges. In the global learning setting, it is critical to exercise sensitivity for teachers whose experiences differ from those of their students. In addition, this initiative should be expanded into the learning capacity with creative thinking and critical reflection, so that the students need to absorb both knowledge and skills in raising their experiences [24]. It is necessary to be committed to avoiding challenging stereotypes among peers and adults. The sense of openness and cultural humility is required to integrate cultural awareness with values inculcated into the outcomes for student learning [25]. As a result, the commitment of learners in defining their own identities via their social engagement would enhance their responsive awareness and development.

The learning instruction in the multicultural classroom needs to examine indicators of competence to see how well they align with assessing the quality of multicultural skills [26]. This mutual configuration is part of standard assessment processes using the best measures available in the educational program. It is also especially important in relation to social, emotional, academic and school climate. In particular, further attempts to enhance accountability and acceptability are required to strategically integrate professional competence with cultural awareness [27]. The behavioral commitment has to be integrated among all students with different social, emotional and mental abilities that affect their learning within the multicultural environment context.

### 2.3. Methodology Research Design

The approach used in this paper refers to develop the framework of understanding CRCT from the perspective of interracial teachers' partnership commitment. Conducted with a qualitative approach, aimed to use the particular method of the structured interview. This approach has been chosen in enabling the achievement process of obtaining the richness of information and related data [27,28]. The main point is to ensure both specific and comprehensive findings are gained through procedural processes [29]. As a case study with the qualitative research approach, the focus is to explore the model of CRCT from the perspective of interracial teachers' daily school interactions.

### 2.3.1. Participants

This study aims to develop the framework model of CRCT from the perspective of interracial teachers' partnership commitment. The data were collected through qualitative interviews focusing on the significance of a culturally competent teaching performance. Purposive sampling was chosen so that the participants involved in this study could represent the population to be identified, so all the individuals in the group are considered as samples [30]. In order to obtain the particular result in the data coding, the research

question was distributed to ask 10 teachers, labelled as follows: the first Malay male participant is named with the code MM1, while the first Malay female participant is named with the code FM1; moreover, the first non-Malay male participant was named with the code MN1, and the first non-Malay female participant was indicated with FN1.

### 2.3.2. Data Collection

Data collection was conducted among 10 interracial teachers with regards to the interracial teachers' partnership commitment. Through the wording and sequencing of questions, the structured interview arrangements with the schedules performed well in enhancing the credibility and reliability of research data [31]. Through the natural setting for the attributes of qualitative research which can be led, the listed respondents were approached to discuss the issues during the research process. This includes perceiving the behavior inside the setting with face-to-face interaction with the respondent [28]. As a result, the accumulation of information refers to the key instruments spread through talking with respondents through the instruments.

### 2.3.3. Data Analysis

Thematic analysis was employed to obtain the main points of data to give insight into the multicultural school society. Incorporating this approach by conceptualizing the basis of the key consideration of CRCT via interracial teachers' partnership commitment was generated as a way of giving insight into the multicultural classroom. This would enable the development of the framework of understanding and strengthening the actual performance of CRCT to empower the social learning environment in the multicultural school society context.

### 2.3.4. Theme Categorization Procedure

The construction of theme categorization procedure is important in order to determine which themes will emerge and be included as the selected themes from the interview data. It means that the selected themes were built from those which a number of the participants stated and agreed on. For this research, the participants were five Malays and five non-Malays, namely two Chinese and three Indians. Thus, the following table includes Malay and non-Malay categories, which consisted of five participants respectively. As such, the themes will only be constructed if more than half of participants in each category agreed; for instance, one theme with three of participants would be considered and carried forward as a main theme in the discussion. Themes with the agreement of less than half of a category will not considered as built themes. This categorization of built themes was determined by allocating the themes with more participants' statements on it during the interview process. The more participants mentioned a certain theme, the more consideration would be more made for it being a theme.

## 3. Results

### 3.1. Malay Teachers' Partnership Engagement Commitment

Opinions on social interaction mainly with others' different backgrounds could provide a valuable supplement to achieve harmony amidst diversity. The extent one's personal ability to be involved in societal life refers to the possession of certain core components to be measured in the awareness stage. From the Malay teachers' results, the themes that resulted from the interview session are (a) integrity involvement, (b) cooperative responsiveness, (c) mutual understanding, (d) responsive awareness, (e) open-minded engagement and (f) mutual respect. The stage of this association with the partnerships' engagement is a key point in disseminating the way of building social harmony established throughout their life. The details can be referred seen in Table 1.

**Table 1.** Malay teachers' partnership engagement commitment.

| No. | Opinion on Interaction | FM1 | FM2 | FM3 | MM1 | MM2 |
|---|---|---|---|---|---|---|
| 1. | Mutual understanding | X | X | X | X | X |
| 2. | Cooperative responsiveness | X | X | X | X | X |
| 3. | Responsive engagement | X | X | X | X | X |
| 4. | Integrity involvement | X | X | X | X | X |
| 5. | Mutual respect | X | X | X | X | X |
| 6. | Open-minded engagement | X | - | X | X | X |
| 7. | Less respect and cooperative | - | X | X | - | - |
| 8. | Political purpose | - | - | - | X | - |
| 9. | Religious diversity | - | - | - | X | - |
| 10. | Selfish | - | X | - | - | - |
| 11. | Lack of responsiveness | - | X | - | - | - |
| 12. | God involvement | X | - | - | - | - |

### 3.2. Mutual Understanding

Based on the findings, all five Malay participants (FM1; FM2; FM3; MM1; and MM2) pointed out the importance of mutual understanding in their experiential basis in the way of the partnership's engagement in our interactions, as stated by one participant, saying

" . . . *We have no problem because they can accept the Islamic culture and we accept each other. I also do not accept any interference from other religions and they are also well-behaved* . . . " (MM2).

Moreover, the extent to which one is capable of using mutual understanding to look at the circumstances has a pivotal role to enlarge the atmosphere where all may have chance to live in a peace. As such, it is necessary to commit to this so that the outcomes can be strengthened through the educational program for societal life.

### 3.3. Cooperative Responsiveness

In this view, all five participants (FM1; FM2; FM3; MM1; and MM2) show their awareness of the cooperative encouragement in the sense that it is necessary to integrate consciousness regarding social interaction into one's personal abilities and skills. For instance, one participant confirmed,

" . . . *because my intentions are not only educating the Muslim students, but also educating all of the including non-Muslim* . . . " (FM1).

Such cooperative responsiveness is needed to inculcate the awareness stage of giving positive feedback through their experiences and views clearly stated in the above example. Thus, both social responsibility and responsive engagement need to include the awareness the circumstances of social interactions to lead to a harmonious societal life.

### 3.4. Responsive Awareness

Among these participants, giving kindness was noted as a good example: five participants (FM1; FM2; FM3; MM1; and MM2) clearly stated that their partnerships' different backgrounds means they are aware of their behavioral relevance in social interactions. Based on these findings, for example, one participant's view was

"*first of all we can still keep the friendship again even though they have not met long ago, but are still in touch and share stories*" (MM1).



The responsiveness here refers to the attempts in understanding the social interaction in the partnership for those different backgrounds. This may give insights into one's societal life circumstances.

### 3.5. Integrity Involvement

As pointed out for all five Malay participants (FM1; FM2; FM3; MM1; and MM2), the subsequent point of their response to the partnerships is related to the involvement of integrity. For example, one of the participants' views could be seen in the following quote.

*"about my opinion, while studying, I think they are very selfish (with laughing) . . . first time, we are all overweight; there is no sense to share with others. But we are also selfish, we feel inferior"* (IIFM2).

It seems that her views could impact her personal experiences, which manifests in certain behaviors, due to the lack of active participation in teamwork in the program. Such a situation could shift to a good outcome as long as there is an involvement of integrity, i.e., commitment to one's work in to manage their teamwork circumstances to achieve a good societal life condition. As stated by a similar participant,

*"our relationships are getting closer and friendly. If there is no such activity, our relationship will be far away"* (FM2).

Moreover, the following point gave a view of the involvement of integrity in participant closeness, such as making jokes, to create circumstances to become more closely related even with multiracial individuality, as underlined here:

*" . . . Yes. It means students are interested in, when we are close to people. Similarly with the teacher, I like to talk to anyone in the form of jokes . . . Of course because this is all my assets . . . "* (MM1).

### 3.6. Mutual Respect

According to all five Malay participants (FM1; FM2; FM3; MM1; and MM2), the subsequent main point of their response towards their partnership is the extent of mutual respect. The mutual respect to engage in interacting with others' different background refers to behaving with the similar feeling regarding their admiration for the other person. For example, the following participant stated.

*" . . . They always respect and understand the religion of Islam . . . "* (FM3).

Their views show the necessity of appreciating one's abilities and qualities in social interactions in order to lead to better societal circumstances, mainly among their partnerships, in the sense that it implies a way of treating others that reflects how one thinks about someone else.

### 3.7. Open-Minded Engagement

As revealed by four of the Malay participants (FM1; FM3; MM1; and MM2), they are aware of with the importance of open mindedness. For instance, one participant clearly pointed out his experience of social interaction with his peers:

*" . . . they are open-minded and that they looked more respectful . . . I will ask them to be patient because God knows . . . "* (FM1).

The view towards open-mindedness refers to feeling conscious of displaying an open attitude, enabling them have a good relationship within the societal life at large. Thus, such engagement is a basic point of view to enhance the situational experience to build this open feeling to disseminate it among others in good circumstances.

### 3.8. Chinese and Indian Teachers' Partnership Engagement Commitment

In this section, from Chinese and Indian teachers' point of view, the themes resulting from the interview session were as follows: (a) open-minded skills, (b) respectful engage-

ment, (c) responsive awareness, (d) tolerance sincerity, (e) professional integrity, and (f) mutual assistance. The stage of this association with the partnerships' engagement is a key point in disseminating the way to build social harmony that will be established throughout their life. The details can be referred to in Table 2.

**Table 2.** Indian and Chinese teachers' partnership engagement commitment.

| No. | Opinion on Interaction | FN1 | FN2 | FN3 | FN4 | MN1 |
|-----|------------------------|-----|-----|-----|-----|-----|
| 1. | Tolerance sincerity | X | X | X | X | X |
| 2. | Respect engagement | X | X | X | X | X |
| 3. | Responsive awareness | X | X | X | X | X |
| 4. | Open-minded skills | X | X | X | X | X |
| 5. | Professional integrity | - | X | X | X | X |
| 6. | Mutual assistance | X | - | X | - | X |

*3.9. Tolerance Sincerity*

As detailed by all five participants (FN1; FN2; FN3; MN1; and MN2), a key point underlying their opinions on social interactions is tolerance sincerity, which refers to engaging in the acceptance of the differences between people in a behaviorally sincere manner. For instance, here is an empirical example:

" ... *If there are things that are less in line with the Muslim tradition or culture they are not angry with me. They can be said to be too tolerant to me and can accept my way* ... " (FN2).

When displaying tolerance to community activities, emphasizing skills and behavioral awareness, one needs to engage in this tolerance willingly and genuinely build social harmony.

*3.10. Respect Engagement*

The next view towards the opinions of interactions, as detailed by all five participants (FN1; FN2; FN3; MN1; and MN2), regards the personal quality of respect when engaging in mutual acceptance in a social interaction. An empirical example can be seen in the following:

" ... *Actually I'm a lonely person so I always use my left hand to give them goods and no one is reproaching or angry with me* ... " (FN2).

With the participants' views on determining the necessary actions for how to manage the conditions of mutual admiration, the personal quality of moral manners, with social abilities and qualities, can lead to good social outcomes.

*3.11. Responsive Awareness*

The next point determined to be one of the themes on the opinion of social interactions refers to the responsive awareness, as indicated by all five participants (FN1; FN2; FN3; MN1; and MN2). One of the participants confirmed that responsive awareness, meaning engagement taking into account the conditions imposed by different backgrounds, is very important, as stated in the following:

" ... *While living in a hostel while studying first, my friend of my home, a Muslim who wants to pray, I will be silent and sit still as a sign of respect to them* ... " (FN1).

The responsive awareness as stated in the above example points out that attempts understand a way of interacting with others' different backgrounds can foster good social relationships.

*3.12. Open-Minded Skills*

From the empirical findings, as indicated by all five participants (FN1; FN2; FN3; MN1; and MN2), the next point was on how open mindedness underlies the extent of personal quality in a relationship. For example, one participant underlined:

" . . . *Based on my experience as long as I'm friends with them, they are not so urgent, can share and can joke together* . . . " (FN4).

Being open with others' different backgrounds means that the relationship quality is improved by open mindedness as it enhances one's knowledge and understanding, enabling them to respond wisely to the issues of difference. The insightful value of open-mindedness could be viewed as being the way they respond into others with a conscious attitude to be open, enabling them to have good relationships within society at large.

*3.13. Professional Integrity*

As pointed out by four participants (FN2; FN3; MN1; and MN2), the next point in looking at the feeling of association within partnerships is professional integrity awareness. This attempts to align the professional engagement with moral integrity. An empirical example can be seen in the following:

" . . . *I and they are friends and can accept each other. There are just a few things that cannot be done and cannot be eaten* . . . " (FN3).

The orientation of professional integrity in the commitment to interact and to collaborate with each other should contribute towards the success of acceptance.

*3.14. Mutual Assistance*

The first theme created from the empirical findings on the participants' associations with partnerships is mutual assistance. As indicated by three participants (FN1; FN3; and MN1), the commitment to attempt to enhance the mutual assistance here refers to the incorporation of being involved with other people as an attempt to build harmony in a diverse society. For example, the following participant underlined.

" . . . *I know all those things. If there is activity together, the matter will be taken care of* . . . " (FN3).

As core elements to achieve this goal, strengthening mutual assistance through the certain project assignments could give insights into supporting an effort to sustainably expand and continue mutual purpose.

**4. Analysis and Discussion**

*4.1. Empowering CRCT through Intercultural Teachers' Partnership Commitment*

In line with manifesting this model, it is required to highlight the empowerment of CRCT in creating the knowledge and skills that teachers must balance due to high-stakes accountability. Moreover, an effort to meet the needs of diverse students in the 21st-century classroom requires a mutual engagement in presenting the focus of intercultural partnerships [31]. Bringing classrooms with such experiences to support academic learning requires the students to possess powerful cultural knowledge. This knowledge is associated with multilingual literacies to guide the students to achieve social and responsive citizenship [18]. In the practical aspect, with such potential, this conceptual framework could be taken in further among diverse learners [32].

The way to engage in educating the students' cultural integrity including language and lived experiences should do so by constructing trusting and caring relationship. This would be a good task to reflect upon among teachers for transferring both knowledge and such skills to come up with positive influences [33]. Associated with bringing CRCT into the educational system, cultural diversity engaged in students' experiences needs to empower them by making them aware of enhancing the successful achievement in the academic field. The phase could be viewed on such the following Figure 1 below.

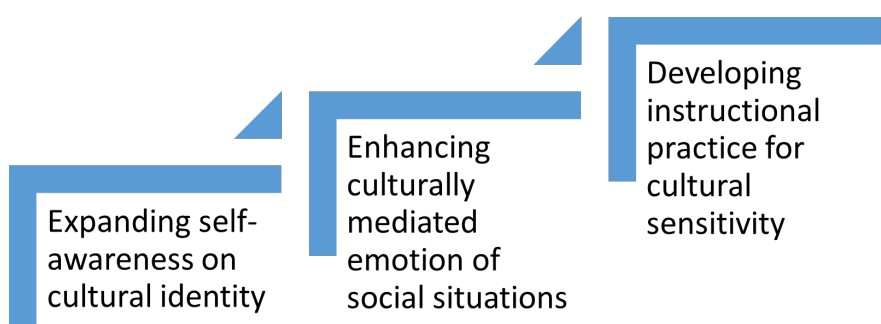

**Figure 1.** Empowering CRCT through intercultural teachers' partnership commitment.

Strategic attempts to demonstrate the impact of CRCT among the teachers' and students' relationships has to do entirely with the way of comprehending the extent of students' cultural, linguistic, and lived experiences. The ultimate aim is concerned with fostering students to be successful, not only academically but also in terms of personal and social quality [34]. As a result, it is considered as a way to educate the students to be aware of other cultures, languages, and lived experiences. Moreover, the way of understanding life phenomena surroundings in terms of school and society should be built in further by trusting and caring relationships [35]. The teaching style has to be integrated with cultural awareness in enabling the knowledge acquisition process in school communities. CRCT is necessary to accommodate a comprehensive understanding of these actions to influence the teaching methods and lessons for planning development [36]. Good impact would be reached from teaching and learning styles associated with CRCT in the public school system. With this regard, the students' engagement in cultural diversity needs to be addressed and set out in the school system.

In addition, the engagement process of cultural diversity among the students could be reflected in the societal transformation pathway. Providing the chance to develop knowledge, skills, and values with a central characteristic would foster social and critical agency [37]. As a result, those who make self-reflective decisions to implement an effective method could do so through personal, social, political, and economic actions. The mutual alignment of both knowledge and skills on intercultural awareness could be developed in implementing further notions of CRCT among active citizens.

In terms of the students' learning, promoting the power balance with equality and justice needs to engage in transforming the experiential learning in the educational system [38]. The effort to have a look into both personal and social aspect on being an active citizenship should relate to building critical skills among the social community. As such, it is a good opportunity for school-community projects to prepare students in reflecting on ways to take action in solving such challenges faced by community members. In this regard, CRCT needs to be regulated to prepare the teachers for accommodating the students' way of learning on behalf of transforming the multicultural values in a pedagogical style to support them in advocating their school.

*4.2. Expanding Self-Awareness on Cultural Identity*

The way of preparing the self-awareness among the teachers aims to enable them to critically reflect their cultural identity in their teaching performance. Their teaching approach might provide cultural awareness in the students' learning in order to provide an effective instruction to transform it in a multicultural classroom [39]. In this regard, the cultural awareness in the social dynamics requires learning styles with life experiences engaged with teaching effectively. As a result, enabling educators to be aware of social and cultural awareness itself should provide a way of putting forward these frameworks appropriately in relationship styles [40]. The strategic phase included the family discipline strategies and views of time and space, particularly in the teaching approach. An attempt of the teaching approach can be implemented in referring to the knowledge basis for culturally responsive teaching [41].

Making conscious effort to get to know the students' activities is conceived alongside interpreting the students' behavior. This alignment refers to the idea that an understanding of students' cultural norms should go beyond, into specific practices of awareness, to reflect upon the learning needs of diverse students [42]. For instance, teachers need to construct the course content around the culture of respect among student diversity, in order to strengthen the students in using culturally representative texts in classroom instruction. In the midst of the community and the classroom context, this enables the achievement of equity in education being more reflective in contributing the perspective of culturally diverse backgrounds. The strategic practice of expanding self-awareness on cultural identity could be clearly viewed into the following Figure 2.

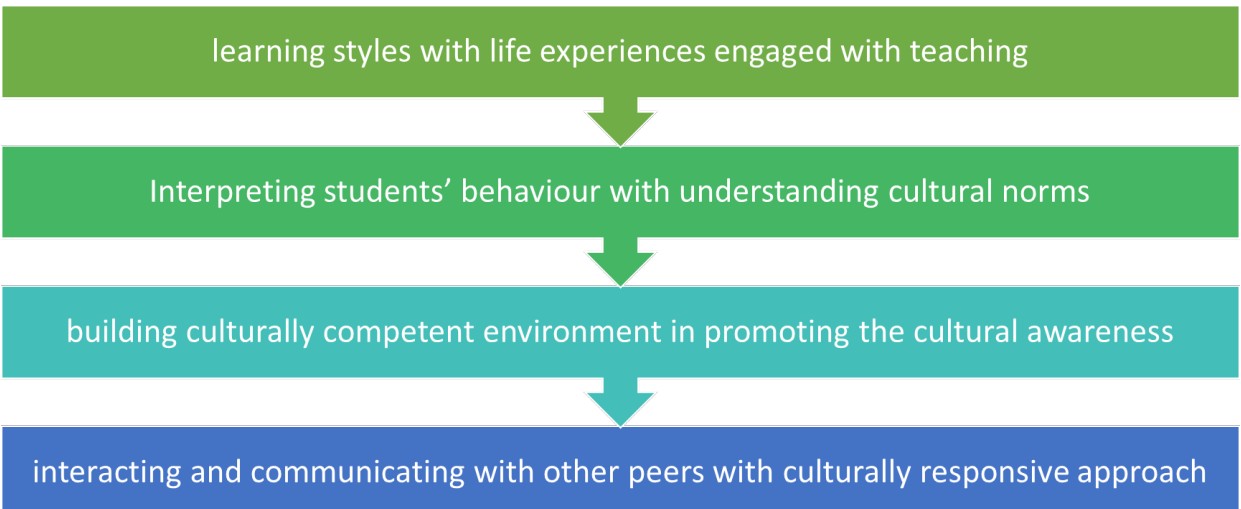

**Figure 2.** Expanding self-awareness on cultural identity.

In addition, it is important to maintain behavioral commitment reflected in the cultural environment and the multicultural classroom. An attempt to build a culturally competent environment should be incorporated in considerations of the classroom and community contexts, enabling the students to help each other [43]. In order to design this in practice, further elaboration could be formulated into policy at the institutional and individual levels. As such, it needs to focus on helping the students' learning ability to strengthen their identity positively on the students' potential outcomes [44]. From this view, making benefit through academic achievement should be resolved into direct implications among the diverse students. This includes how and what students in diverse classrooms could be taught about multicultural education, addressing their diverse backgrounds [45].

In order to emphasize the cultural competency approach, an attempt to implement this in the midst of diverse backgrounds should be carried out at the institutional level. This refers to an asset to learning the cultural considerations by keeping in mind on mindset [46]. In particular, enhancing the policy development through classroom practices and interactions with stakeholders is necessary to reform the school-wide cultural competency in premising cultural particularities. Acquiring detailed and factual information in supporting diverse students to grow their cultural competency at the classroom level has an appropriate look at promoting the cultural awareness combined with inclusiveness towards culturally competent schools [47].

Further, the curriculum design is prioritized to go a long way in fostering the accommodation of rules and regulations, with a keen interest in learning to other cultures. As a result, it can be facilitated in a multicultural classroom to allow diverse students, through their attitudes in interacting and communicating with other peers, to expand their knowledge. This encourages teachers to demonstrate and teach the skills to enhance students' proximity to their peers from other cultures [48]. It ensures the role of communicative approaches as the initiative to strengthen contacts with their peers. Such an approach

would enhance the positive and empathetic values essential to demonstrate how to teach diverse students. From this view, it is an opportunity to make students aware of their peers from other cultures.

To give a picture with a broader perspective, designing the approach of CRCT aims at shaping the learners' knowledge focusing on physical and mental skills in the multicultural classroom. This curriculum instruction designed into modules and programs with considerable detail has its desired effect in producing the stated intended outcomes for all students [49]. The specific quality and efficacy of these experiences through the choice of course experiences within teaching become fundamental to the quality of any curriculum. It is essential to create an effective teaching framework with the improvement of curricular quality to make it clear and transparent to the students across deliberating kinship in the midst of the multicultural classroom. The following phase of learning outcomes is integrated into the methods and the learning activities. The extensive point of the way to assess and design in enabling diverse students aims to achieve the intended learning outcomes. As such, it is expected to ensure the culturally responsive approach with communicative teaching.

### 4.3. Enhancing Culturally Mediated Emotion of Social Situations

Culturally mediated emotion is regulated into teaching instruction and characterized in social situations for learning processes. The current form of its usage is applied in culturally appropriate and culturally valued knowledge in the curriculum content [50]. From this point of view, an attempt to transform the content of valued knowledge into the curriculum design needs to reconstruct the way that makes students develop and solve chances and challenges with knowledge and skills. This needs to be integrated wisely into the communicative teaching model with a culturally responsive approach in the curriculum context.

In addition, the extent to which the multicultural classroom and curriculum is assigned in developing culturally responsive emotions amongst the students' diverse cultural backgrounds ranges [51]. Attempts to enhance communicative methods could be mediated to the social situation. As a result, the effective way to integrate the recognizing of the uniqueness of cultures, languages, and communities would enable the teachers to enhance cultural and linguistic diversity in the multicultural classroom [52]. It is required to have a sufficient awareness of social identities and cultural emotion in the multicultural classroom. In this regard, enhancing the entire initiative to work together with commitment and a good attitude is actually an outstanding value in fostering the students' unique strengths to create equitable classroom communities. The detailed attempts to enhance culturally mediated emotion of creating social situation are governed with the balance between content and context of teaching practice enhancement as noted in the following Figure 3.

In addition, teachers are encouraged to have a social skill commitment in interacting with all learners without exception. The strategy for this is determined towards enabling the individuals with diverse backgrounds to cooperate together peacefully [53]. Initiatives to foster socialization among students from different background in a multicultural classroom are important; they would enable the students to have equal rights in a variety of educational experiences. The core points on enhancing culturally mediated emotion of social situation are addressed as noted in the Figure 4 above. The potential value aims to help them routinely use various techniques in correlating their practice to solidify their learning with socially responsive and responsible engagement [54]. For instance, having an active role and participation in language, literacy, and life is required to prepare well for recognizing and incorporating both knowledge and experience into classroom practice. In this view, culturally defined identities would become valuable among the students in bringing the fruit of their knowledge into their learning communities [55]. It is important to note that mainstream power discourse needs to be inculcated among the diverse students to allow them to be willing to cross personal and professional boundaries in pursuit of social justice and equity.

# Enhancing Culturally Mediated Emotion of Social Situations

Transforming the content of the valued knowledge into the curriculum design needs

Reconstructing the way that makes students develop and solve chances and challenges with knowledge and skills

The effective way to integrate in recognizing the uniqueness of cultures, languages and communities

Enabling teachers to enhance cultural and linguistic diversity in multicultural classroom

Enhancing the entire initiative to work together with respectful attitude

An outstanding value in fostering students' unique to strengths to create equitable classroom communities

Enabling the students to have the equal right on the variety of educational experiences

Observing, learning and emulating adult behaviour and modelling as a guide in preparing their future life coexistence

Demonstrating how to teach diverse students as an opportunity to make aware towards their peers from other cultures

Creating an effective instruction to foster the students across deliberating kindship in the midst of multicultural classroom

**Figure 3.** Detailed attempts on enhancing culturally mediated emotion of social situations.

Characterizing usage on culturally appropriate social situations

Learning practice with socially responsive and responsible engagement

Posessing initiative to link other students from the different background

**Figure 4.** Core points on enhancing culturally mediated emotion of social situations.

Therefore, it is necessary to integrate the curriculum design to allow the teaching style to be meaningful in advocating models of social justice and equity to prepare in ordering knowledge and skills [56]. It is an attempt to utilize a variety of learning strategies, such as cooperative learning connected to real life, centered within the diverse learning styles. As such, the strategic coordination with having the commitment to be aware of the cultural factors might give an influencing factor of the students' performance [57]. It is the model acceptance of diversity and student variance to observe, learn and emulate adult behavior and modelling as a guide in preparing their future life coexistence.

## 4.4. Developing Instructional Practice for Cultural Sensitivity

Attempts to develop the cultural sensitivity in the classroom community should pay attention to elaborating the teachers to develop the learning activities significantly relevant to their students' cultural experiences. The students are encouraged to stretch beyond the range of organizing the cultural sensitivity. Teachers have at their command a range of teaching approaches to reach students in various culturally appropriate ways [58]. Both

language and culture, which should belong among the students, can be used as a tool to encourage their awareness of social and individual identity. Giving feedback to enable them to share with the students has to look into cultural identity awareness [59]. As such, looking for a comprehensive understanding might start with examining the environmental community and social responsibility. It is important to keep maintaining the comprehension pathway about the roles to be played continually for effective classroom management [60]. This would enable the teachers to develop procedures and routines in working. The extent of having such confidence in classmates can be enhanced among students in developing cultural skills with social feeling. The initiative on developing the instructional practice to achieve the cultural sensitivity is clearly addressed in the following Figure 5 below.

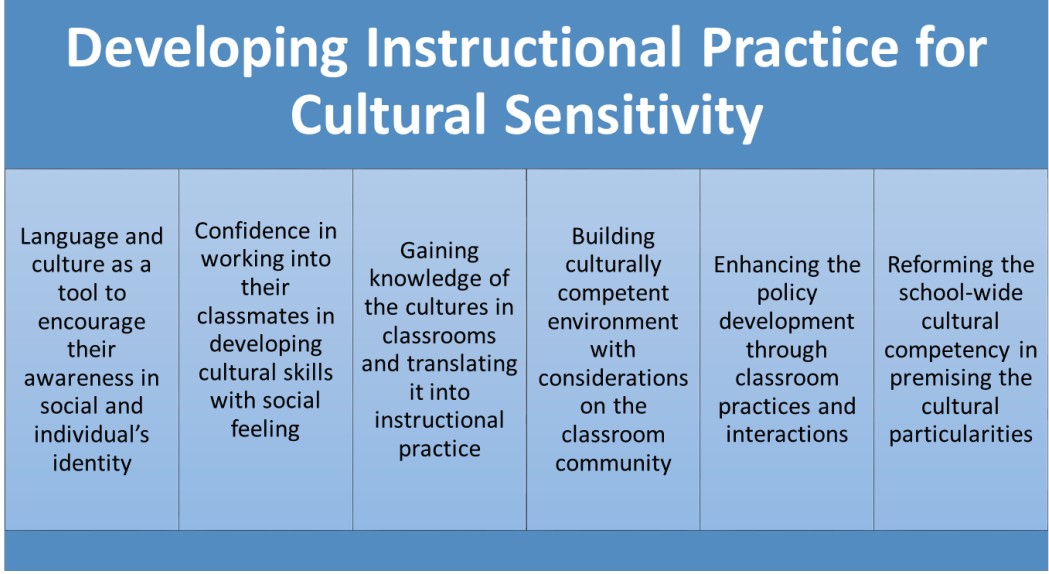

**Figure 5.** Developing instructional practice for cultural sensitivity.

In addition, enhancing the learning opportunities to engage with language and culture, for instance, would give the advantage of experiencing instructional skills in practice. The particular essence of cultural sensitivity refers to the closer correspondence in enabling the higher education institution to carry out in the learning process [61]. The way in which students need to go further in the learning process refers to core identities, one of which is cultural identity, across the multicultural classroom where such experiences become engaged to maximize learning opportunities. As a result, teachers are required to gain knowledge of the cultures represented in their classrooms and translate this knowledge into instructional practice [62]. It indicates that this instruction with such cultural sensitivity can be viewed to recognize students' diversity in class.

Further, the main concern is solely determined to focus on easily stereotyped artefacts of culture, such as food and art. This particular contribution could be cultivated into building the cultural sensitivity beyond interpreting the students' behaviors within the cultural context [63]. Towards an ultimate view that teachers need to understand the cultures, reflecting on the instructional implications brought about by this cultural knowledge can be represented in their classrooms. Developing learning tasks among the students would become meaningful in underlying culturally mediated emotion [64–66]. As such, it has to do with the ways of knowing, understanding, and representing information of cultural identity. It seems similar with the culture of each student where the cultural sensitivity is valued to underlie the knowledge on how to do so in the ways of communicating and learning [67]. Instead, students are seen as having knowledge and experiences with the inclusion of knowledge that is relevant to the student's life.

### 4.5. Implication for Experiential Learning in the Multicultural Classroom

The implications of this study could be viewed as the following. The strategic approach of CRCT in a multicultural perspective is required to possess such steps among the teachers in a multicultural classroom [68]. Attempts to enhance the critical element in bringing a variety of aspects such as events, concepts, issues, and themes from multiple perspectives are integrated in their experiential learning. In this regard, these should be widely spread to allow students to explore the multicultural aspects of the curriculum offered in higher education. Reflecting everyday aspects of living and students' daily experiences is considered a necessity in creating analogies related to the selective description among teachers [69]. As a result, the need to develop knowledge about sociocultural backgrounds has to be gathered by including the students' experiences. This would enable the sources used to be represented as frequently as possible, in an approach to help students understand certain events viewed in diverse ways [70].

Consequently, understanding the relevance of the lives of students in reflecting upon the images of natural experiences should be integrated into curriculum design [71]. It is important to communicate the depth of meaning when recommending that teachers should prepare such subject matters for the students to develop their critical skills. To reflect upon living via daily life experiences, it should begin by challenging oppression in society. It would be significant to see the necessity where the students should be given an accurate, well-rounded view on their accomplishments shared among the students [72]. In this view, such aspects including social, economic, political, and geographical conditions in existence at certain places and times would give feedback in influencing the way of dressing, eating habits, and other customs of certain people [73]. This would make students aware of the experiential substance in the society. With this regard, it can be considered to underlie the significance of today's society to support the experiential learning among students. The detailed point of how the CRCT is addressed to give a value on teaching practice amidst the multicultural context could be referred to the following Figure 6 below. This indicated that the need to expand individual ability to provide an appropriate support should be consolidated in an effective way to actively engage with the diverse community.

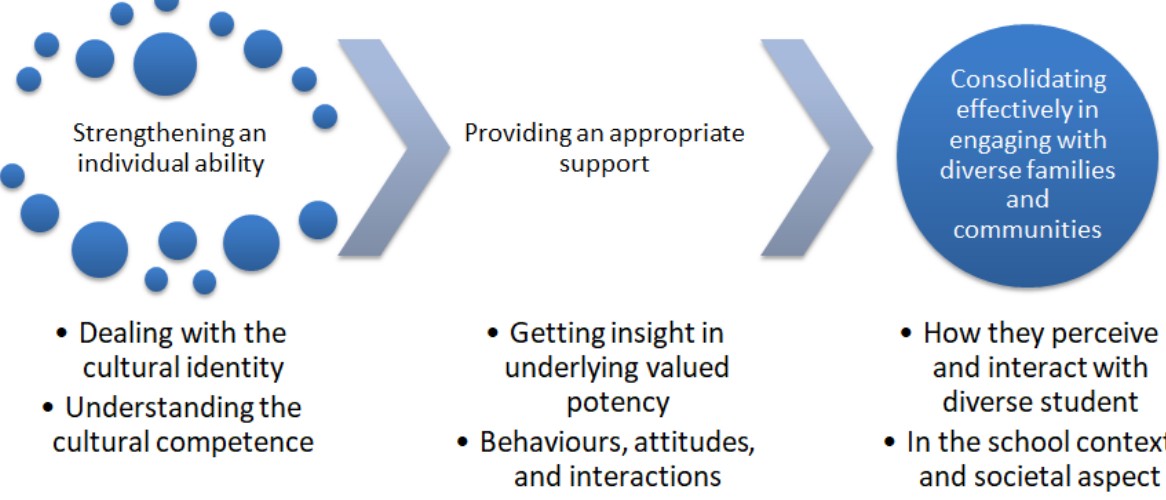

**Figure 6.** Implication for experiential learning in the multicultural classroom.

In addition, the main focus of the integration of content and context across disciplines in enabling the learners' sufficient understanding should be determined in the multicultural classroom. As a result, the students need to understand that all things in life are interconnected in a way that can make many activities lively accustomed in their daily lives [7]. This should be widely regulated into the teaching in a compassionate approach. The particular concern is incorporated into the classroom environment by demonstrating the value of diversity among students and educators [74]. It indicates that the instructional

design, activities, interaction patterns, behaviors, and expectations are necessary to make things fair and equitable for all. The educators need to be keenly aware that many of the traditional school patterns accommodate students and work consistently [75,76]. As such, the interaction patterns among some students' learned communication style has to do consequently with something which is acceptable and relevant to the diversity issue.

Towards the pluralistic society, it is required for some students to think of responding to a challenging issue in order to help them answer questions indirectly. Attempts to give extraneous information are structured to strengthen the students' mobility in the multicultural classroom [77]. Such a process among the elements, including classroom organization, aims to strategize an effort to promote the teacher's and student's relationships. As a result, it is necessary to examine the use of communication in reflecting images of all the students to focus on their active involvement [78]. With this regard, the sufficient support in leading to the successful attainment among diverse student populations is required. As such, the educators need to critically underline their own assumptions of what schools and classrooms are supposed to in the way that can cover the overall area of classroom climate and needs.

Further, attempting to provide solutions among challenging factors is a main concern in higher education climates. This is because preparing the students' academic success is consequently a beneficial value in enabling the students to develop their critical literacy skills through multicultural pedagogy [79]. In this regard, schools can help students to ensure that they possess citizenship skills to resolve issues engaged with in the school community. They are regarded as active citizens who want access to resources in gaining the benefit for the academic practices inside and outside of the school context [80]. In addition, legitimizing the cultural identity needs to investigate the engagement of experience in the way that pre-service teachers can be carried out.

The strategic approach to focusing on the aspects of the professional practice aims to solve such challenging issues. As a result, the teaching performance is required to possess some elements related to the multicultural classroom, including student readiness, interest and learning profile [81]. In particular among those is the content referring to what the student needs to learn or how the student will obtain access to the information. The commitment to work while considering the activities aims to enable the student to engage with and master the content. It is certain that the process would underlie the extent of outcomes which will create an opportunity to ask the student to demonstrate what they have learned [82]. In particular, the learning environment would also become a way that the classroom can work in feeling personal development via the exposure to linguistically and culturally different professional settings.

## 5. Conclusions

In responding to the challenges of diversity, the strategic approach of teaching in the multicultural classroom is required to empower responsive and communicative skills. This is important to underlie the extent of teaching performance to contribute a valuable essence to emphasize the way that can be carried out in the context of the multicultural school society. An attempt to empower CRCT into diverse learners plays a significant role in contouring the main foundation of how to integrate skills in a comprehensive coverage of theory and applications. This study presents a comprehensive picture of CRCT practices to give examples of some of these practices. This paper has explored the significant essence of CRCT, which has several points in a comprehensive basis, addressing awareness, attitude, knowledge and skills. This would be the core element of making learners aware of reacting and communicating with others of different backgrounds. These are: embedding self-awareness of cultural competence in multicultural classroom, constructing emotional and social development of cultural awareness, and internalizing responsive awareness of social engagement in global learning. All these refer to the core guideline to give a clear understanding and insightful view on the importance of cultural and responsive awareness in the multicultural classroom. The positive feedback could be deployed as a

contribution to empower both ability and skill in solving the challenges in the multicultural classroom. The strategic guideline could be implemented in the interracial school context. The proposed model here refers to the reference which becomes the contribution among those education sectors. These include: students, educators, principals, learning facilitators, researchers, educational technologists, instructional designers, practitioners, and researchers. In particular, those who are interested in exploring, understanding and developing discerning perspectives on diversity and diverse learners in 21st century education are included. Moreover, the strategic approach of CRCT is positively determined to solve the problems faced by the multicultural classroom in the interracial school society context. As such, it should be considered to expect to contribute an insight to provide a particular guideline to give a clear understanding and insightful view into the importance of CRCT for diverse learners. As a result, this teaching approach should be engaged with in an attempt to lead to the significant engagement of diverse learners. This is because it would help to foster critical thinking and social awareness to underlie the implications for students' experiential learning in the multicultural classroom.

**Author Contributions:** Conceptualization, Z.B. and M.H.; methodology, Z.B. and M.H.; investigation, Z.B.; resources, Z.B.; writing—original draft preparation, M.H.; writing—review and editing, M.H.; funding acquisition, Z.B. and M.H. All authors have read and agreed to the published version of the manuscript.

**Funding:** This research was supported by University Research Grant for Special Interest Group (GPUSIG), Universiti Pendidikan Sultan Idris Malaysia, with the reference number 2020-0148-106-01, where Dr. Miftachul Huda is the principal investigator.

**Data Availability Statement:** Not applicable.

**Conflicts of Interest:** The authors declare no conflict of interest.

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
