# Peer review of "Reflecting Culturally Responsive and Communicative Teaching (CRCT) through Partnership Commitment"

_education, doi:10.3390/educsci12050295_

Round 1
Reviewer 1 Report
The topic is relevant and the findings are interesting for the scientific community: the significant essence of CRCT is explored, and several points are finded: a comprehensive basis of addressing awareness, attitude, knowledge and skills. The research is correctly conducted, but the methodology should be better explained. It is well-referencenced, but the final list of references is not correct: there are some unnecessary bold letters and incorrec signs (:).
Also there are some problems with the figures. The numeration have errata and some figures' content should be replaced by text-lists (figs. 3.1, 4...).
Language uses must be improved (monotony, errata) and an homogene dialect should be chosen. There are more marks at the attached manuscript.

Author Response
We are grateful for the positive and constructive comments that originated for the review process. We have responded to the points raised by the reviewers and the editorial comments as follows. The methodology has been revised with an analytical approach developed in the paper, and also the references have been updated. The writing issue has been resolved with proofreading.
Reviewer 2 Report
Dear authors
The following changes are suggested to improve the transparency and comprehensibility of your article
The methodology must be explained in a more transparent and procedural way, not just conceptually. Especially, a table that explicitly links the objective with the categories (be they precategories or emerging categories) would be very necessary.
Given the appearance of the results in categories that emerge without any known logical basis, it is necessary to explain the basis of these categories.
It is suggested to separate: Analysis-Discussion
Regards
Author Response
We have responded to the points raised by the reviewers and the editorial comments as follows. The methodology has been revised with an analytical approach developed in the paper, and also the references have been updated. The writing issue has been resolved with proofreading. We are grateful for the positive and constructive comments that originated for the review process.
We do hope that the revision made might reach the reviewers’ positive and encouraging comments. Therefore, such helpful suggestion and advice has been already conducted in the latest version of our manuscript. Hopefully, this revised manuscript can reach a standard as supposed.
Reviewer 3 Report
Dear authors,
Congratulations on the manuscript! It is a very interesting topic.
As suggestions:
- it would be advisable to provide more details on how to organize the research (selection of participants, duration of discussions, transcription, the confidentiality of data, etc.);
- please reread the way to organize bibliographic references in the text and at the end of the text.
Best regards,
Author Response
Many thanks for the constructive comments. The paper has been revised in following scientific basis. The analytical approach has been developed in the paper, and also the references have been updated. The writing issue has been resolved with proofreading.
We do hope that the revision made might reach the reviewers’ positive and encouraging comments. Therefore, such helpful suggestion and advice has been already conducted in the latest version of our manuscript. Hopefully, this revised manuscript can reach a standard as supposed.
Round 2
Reviewer 2 Report
Dear authors
The pop-up appearance in the category results is still a lighting species that has not been enriched since the previous version. It is not questioned that it is contingent: but what is its foundation? Do they emerge only from empirical data? There is no traceability of the origin of that categorization. This could be corrected by incorporating a subsection on methodology that bases this emerging categorization, and that the methodology is not only a conceptual theoretical approach. The qualitative methodology cannot be an argument for the proposal of categories to be an emerging process without a logical link.
Greetings
Author Response
Response to Reviewer 1 Comments
Point 1: The pop-up appearance in the category results is still a lighting species that has not been enriched since the previous version. It is not questioned that it is contingent: but what is its foundation? Do they emerge only from empirical data? There is no traceability of the origin of that categorization. This could be corrected by incorporating a subsection on methodology that bases this emerging categorization, and that the methodology is not only a conceptual theoretical approach. The qualitative methodology cannot be an argument for the proposal of categories to be an emerging process without a logical link.
Response 1: Dear reviewer. Many thanks for your constructive feedback. The subtheme of categorization of built theme was made with the following consideration. The construction of theme categorization procedure is important in order to result in determining which themes will be emerged and included as the selected themes from the interview data. It means that the selected themes were built from which the number of par-ticipants have stated and agreed on it. On this research, the participant number comes from five Malays and five non-Malays, namely two Chinese and three Indians. From this point, the category of table is based on the Malay and non-Malay, which consisted of five participants respectively. As such, the themes will only be constructed along with more than half of participants in each category, for instance the one theme with three of them would be considered as a theme. So, the theme, which was stated among more than half of them, will be considered as the main theme in the discussion. The one with less than half will not considered as the built theme. This category of built themes was determined through allocating the one theme with more participants’ statement on it during the inter-view process. The more participants stated on the certain theme, the main consideration would be more made on it for being a theme.

This manuscript is a resubmission of an earlier submission. The following is a list of the peer review reports and author responses from that submission.